behaviour/complexity/statistical physics

gender, networks, academia

**Author for correspondence:**
Andreas Orland
e-mail: aorland@uni-potsdam.de

# Is there a gender hiring gap in academic economics? Evidence from a network analysis

## Andreas Orland[1] and Max Padubrin[2]

[1]Department of Economics, University of Potsdam, August-Bebel-Strasse 89, 14482 Potsdam, Germany
[2]Department of Economics, Technical University Berlin, Strasse des 17. Juni 135, 10623 Berlin, Germany

AO, 0000-0001-6954-3669

We collect a network dataset of tenured economics faculty in Austria, Germany and Switzerland. We rank the 100 institutions included with a minimum violation ranking. This ranking is positively and significantly correlated with the Times Higher Education ranking of economics institutions. According to the network ranking, individuals on average go down about 23 ranks from their doctoral institution to their employing institution. While the share of females in our dataset is only 15%, we do not observe a significant gender hiring gap (a difference in rank changes between male and female faculty). We conduct a robustness check with the Handelsblatt and the Times Higher Education ranking. According to these rankings, individuals on average go down only about two ranks. We do not observe a significant gender hiring gap using these two rankings (although the dataset underlying this analysis is small and these estimates are likely to be noisy). Finally, we discuss the limitations of the network ranking in our context.

## 1. Introduction

Various gender gaps have been documented among academic economists. Female economists earn less than their male colleagues [1,2] and are less likely to receive tenure and to be promoted to full professor [3]. The finding that, in economics, male faculty members evaluate (hypothetical) female candidates less favorably for tenure-track assistant professor positions than (both male and female) faculty members from three other fields might be related to this [4]. The fact that the share of females at each stage of career (from graduate school to full professor) decreases is known as the 'leaky pipeline' [5,6]. Higher-ranked

academic institutions (where rank is measured by research productivity) have fewer women in both junior and senior positions than lower-ranked institutions [7]. As economics is both a field with a low share of female faculty and a field with a gender gap in tenure and promotion [8], it is important to identify existing gender gaps.

This paper builds on an approach by Clauset *et al.* [9]. They collected a network dataset of faculty in three different disciplines (business, computer science and history) in the USA and Canada. They then used a minimum violation ranking [10] to calculate a social ranking of the academic institutions, and show that on average, faculty step down 27–47 ranks from their doctoral institution to their employing institution. In addition, they document a significant 'gender hiring gap'—women had to step down further than men.

Our network data consist of tenured economics faculty in Austria, Germany and Switzerland.[1] Besides the variables used in [9], we also account for career age effects. While we also find that faculty step down about 23 ranks on average when we use our network data, we do not find a significant gender hiring gap. This is confirmed when we consider the Handelsblatt and Times Higher Education rankings, even though the number of ranks faculty step down is smaller with these two rankings (where the dataset is smaller than the one underlying the network ranking, thus these estimates are likely to be noisy). The article concludes with a discussion of the limitations of this study, and with a few suggestions for improving data availability and, thus, improving future research on the topic.

## 2. Method

Between 6 April and 5 May 2020, we hand-collected publicly available internet data of all institutions that grant PhDs in economics in Austria (AT), Germany (DE) and Switzerland (CH). All data were independently checked by a student helper and revised.

The institutions are the nodes in our network.[2] An individual faculty member forms the links between the institutions. We consider a directed network: the source is the institution where an individual's PhD was acquired, and the target is the institution where the individual works in a tenured position. Neither individuals with a non-economics PhD (e.g. in related fields like statistics, business, political science or agricultural economics) nor individuals with an economics PhD working in a non-economics department/group (e.g. in one of the aforementioned related fields) are included. Initially, we use a node to account for the individuals that come with a PhD from outside AT/CH/DE into our network. Additional variables are the individual's gender and the year in which the PhD was acquired (which we transform into career age in years).

Table 1 summarizes the data. The full network contains the node with the individuals with a PhD from outside the sample. The share of individuals that acquired their PhD outside AT/CH/DE is 22.7%; Switzerland has the highest share and Germany the lowest. For the closed network, we restrict our sample to the individuals with both their source and target institution in AT/CH/DE. We consider 100 institutions and 552 individuals. The individuals' mean career age is about 20 years and only 14.7% of them are female, with Austria having the largest share and Switzerland the lowest.

The restriction that we use for closing the network raises the question of whether there is a difference in the gender composition of included and excluded individuals. Table 2 shows the numbers and shares of male and female individuals inside and outside the closed network. The share of female individuals inside and outside the closed network is very similar, both when we consider all three countries together and separately. Fisher's exact tests, also reported in table 2, confirm that there is no significant association between the gender composition and included and excluded individuals in all three countries, together

---

[1]We consider these three countries for three reasons: (i) A dataset including three countries supplies us with more observations of individuals and institutions than one single country. (ii) Most of the included faculties are German-speaking (exemptions are the institutions in the French- and Italian-speaking parts of Switzerland). Within Europe's generally flexible labour market, the common language should facilitate the exchange of individuals between the included institutions. (iii) Many associations (like the *Verein für Socialpolitik*, the association of economists in German speaking-countries) and rankings (like the Handelsblatt ranking; it also includes the institutions in the French- and Italian-speaking parts of Switzerland) also focus on these countries' economics faculties, thus making comparisons between rankings easier (and our study potentially interesting for a specific audience). For an analysis of the research and publication activities of economists that received their PhD between 1991 and 2008 in these three countries, see [11].

[2]For an introduction to social and economic networks, see [12].

**Table 1.** Summary statistics of the full and closed network. (Note: the total of the full network includes a single node representing all individuals with a PhD from *outside* AT/CH/DE. The closed network only includes individuals in AT/CH/DE with a PhD from within this closed network. Standard deviations in parentheses.)

| | full network | | | | closed network | | | |
|---|---|---|---|---|---|---|---|---|
| | total | AT | CH | DE | total | AT | CH | DE |
| no. institutions (nodes) | 101 | 8 | 12 | 80 | 100 | 8 | 12 | 80 |
| no. individuals (links) | 714 | 49 | 129 | 536 | 552 | 36 | 72 | 444 |
| mean career age | 19.9 | 18.7 | 20.3 | 19.8 | 20.2 | 18.9 | 21.2 | 20.2 |
| | (8.0) | (7.3) | (7.7) | (8.2) | (8.1) | (7.5) | (7.8) | (8.2) |
| share female | 15.4% | 22.4% | 1 0.1% | 16.0% | 14.7% | 19.4% | 9.7% | 15.1% |
| share PhD outside sample | 22.7% | 26.5% | 44.2% | 17.2% | — | — | — | — |

**Table 2.** Two-way tabulation of individuals' gender and inclusion in the closed network. (Note: row percentages in parentheses.)

| | total | | AT | | CH | | DE | |
|---|---|---|---|---|---|---|---|---|
| | female | male | female | male | female | male | female | male |
| PhD inside cl. network | 81 | 471 | 7 | 29 | 7 | 65 | 67 | 377 |
| | (14.7%) | (85.3%) | (19.4%) | (80.6%) | (9.7%) | (90.3%) | (15.1%) | (84.9%) |
| PhD outside cl. network | 29 | 133 | 4 | 9 | 6 | 51 | 19 | 73 |
| | (17.9%) | (82.1%) | (30.8%) | (69.3%) | (10.5%) | (89.5%) | (20.7%) | (79.3%) |
| Fisher's exact test | $p = 0.323$ | | $p = 0.451$ | | $p = 1.000$ | | $p = 0.211$ | |

and separately. We thus conclude that there is no significant effect on the gender composition in the closed network from excluding individuals with PhDs granted outside the closed network.

Figure 1 visualizes the closed network using the ForceAtlas2 algorithm [13]. In this force-directed visualization, unconnected nodes repulse each other, while links attract the connected nodes. This means that institutions are closer to one another when they (unilaterally or bilaterally) exchange individuals. The exchange of individuals between institutions of different countries is not pronounced as the institutions of each country are clustered. This observation might be in line with reported homophily—the tendency that people interact with others who are like themselves—in many other social networks [14,15]. However, we do not have further data to support this claim.

The network ranking follows the idea that individuals in societies form prestige hierarchies [16] and translates this idea to institutions. The key assumption is that institutions hire individuals to emulate their (more successful) doctoral institutions. The social hierarchy of institutions is endogenously determined by the observed hiring patterns in the network. If the academic job market followed a perfect hierarchy, we would observe no violations against the ranking and no individual would work at an institution with a higher prestige than their source institution (an admittedly strong but simple assumption).

To rank the institutions in our network, we use the algorithm supplied by [9]. Here, we explain how the algorithm works in our specific case (where we used the default settings with bootstrapping). First of all, a $n \times n$ adjacency matrix is set up (with the $n = 100$ PhD granting institutions in the rows and the $n = 100$ employing institutions in the columns; the numbers in the matrix represent the number of individuals 'sent' from the granting institution to the employing institution). Then, all institutions are ranked in decreasing order of the number of individuals working at either one of the institutions in the closed network (the institutions' *out-degree*), and the percentage of individuals who violate the social hierarchy is calculated (equivalent to the share of all individuals in the network that are *below* the diagonal in the adjacency matrix). Then, during the burn-in phase (i.e. before the sampling phase),

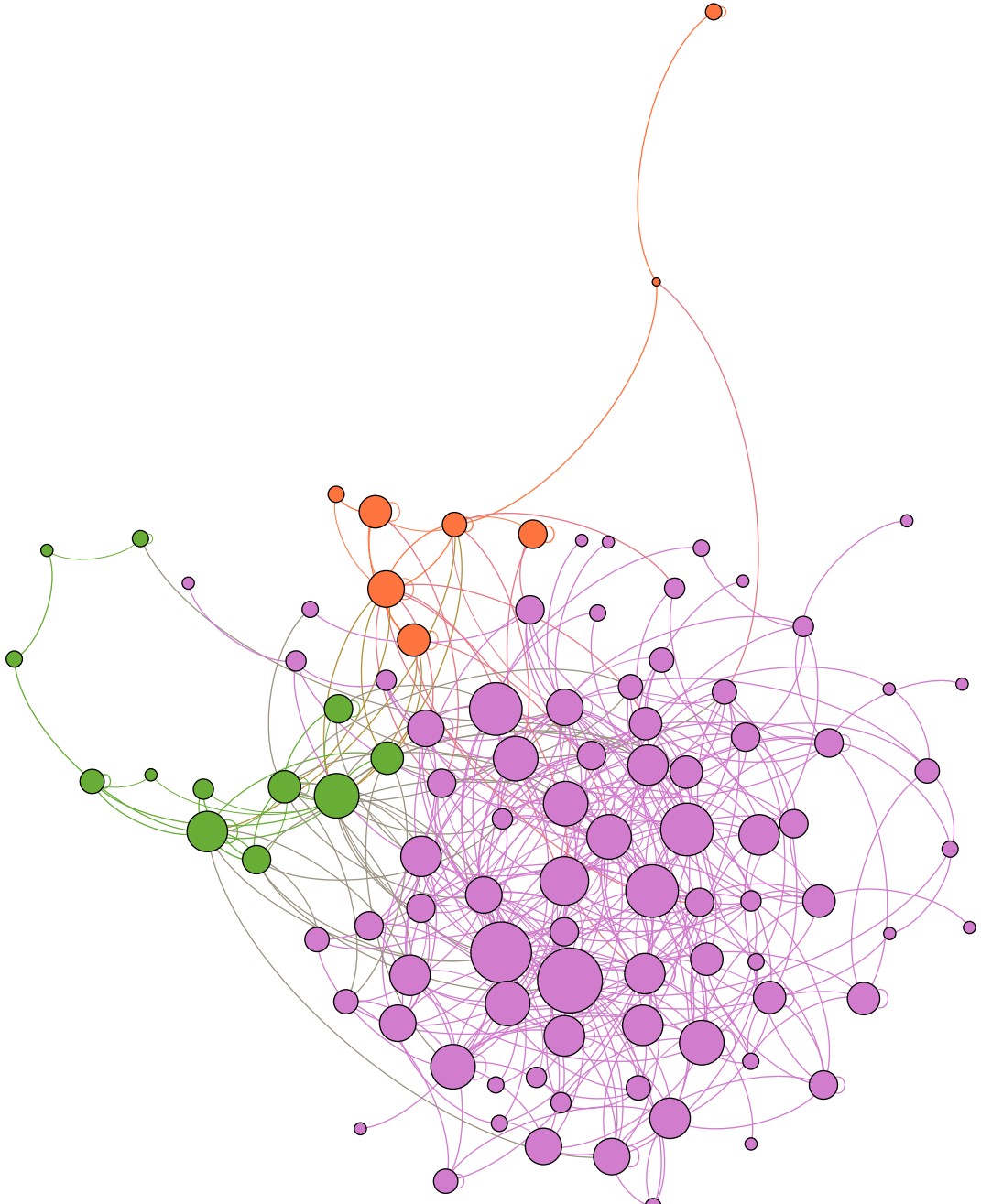

**Figure 1.** A force-directed visualization of the closed network. (Note: Austrian institutions are orange, Swiss institutions in green, German institutions in purple. The size of the nodes corresponds to the number of tenured individuals in the closed network. The size of the links corresponds to the number of exchanged individuals. The visualization is based on the ForceAtlas2 algorithm [13].)

$n \times n = 10\,000$ iterations of the following procedure are repeated: (i) a uniformly random pair of two institutions is chosen and a new ranking is proposed in which their ranks are exchanged. (ii) The new percentage of violating individuals is calculated and compared with the one before the exchange. (iii) If the exchange (non-strictly) lowers the percentage of violations, the new ranking is implemented. Otherwise, the ranking before the exchange is kept.

Thus, during the burn-in phase, in each iteration, the algorithm tries to reduce the number of individuals below the diagonal in the adjacency matrix by sorting them above the diagonal. In the following sampling phase, during $n \times n = 10\,000$ iterations, the algorithm still tries to reduce the number of violations (as in the burn-in phase) and saves the ranking at each of 100 equidistant iterations. The ranking of one repetition of the algorithm is the average ranking of the 100 saved

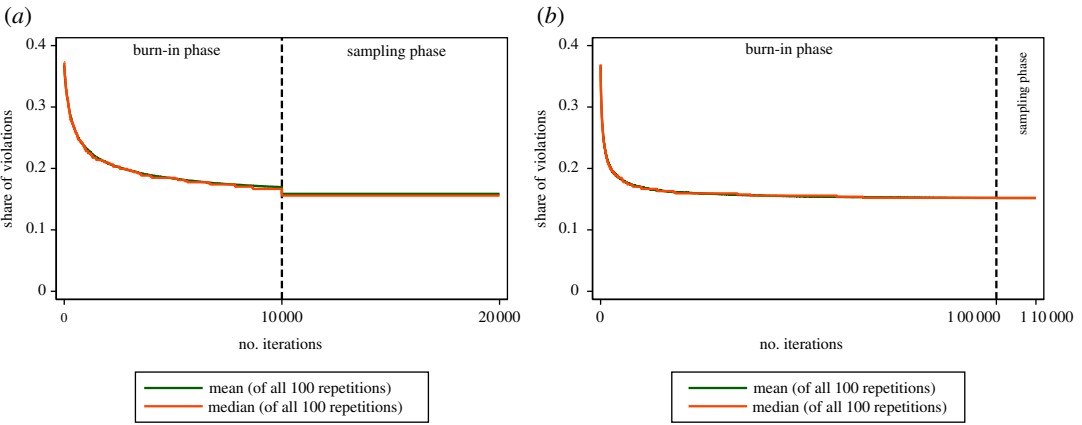

**Figure 2.** Comparison of convergence processes. (*a*) Convergence process of the implemented (default) parametrization with bootstrapping, (*b*) convergence process of the longer-running parametrization with bootstrapping.

rankings during the sampling phase. We used the bootstrapping option to deal with the sparsity of the adjacency matrix:[3] when randomly choosing institutions (with replacement), each institution received a probability equivalent to the share of individuals working in that institution.[4]

Complex networks can produce different rankings that entail the same percentage of violations [17]. Instead of relying on just one (randomly chosen) ranking, we conduct 100 repetitions of the algorithm as stated above and use the average ranking of each institution for our network ranking. Electronic supplementary material, table A1 supplies the rank and average rank of all institutions. We observe only 14.67% violations (i.e. individuals that work at higher-ranked institutions). This number is in line with the findings in [9]: we observe a social hierarchy.

To examine if and how the algorithm converged in our implementation, and also compare our parametrization with a significantly longer-running one. In this longer-running parametrization, we, again, conducted 100 repetitions. This time with a burn-in phase of 100 000 iterations (10 times the number as in our implementation). In figure 2, we compare the convergence processes during the burn-in phase associated with the different numbers of iterations—we show both the mean and the median of all 100 repetitions for all iterations. By comparing figure 2*a*,*b*, we can see that the higher number of iterations does not lead to a lower share of violations. Most of the reduction of the violations takes place during the early iterations of the algorithm. We thus conclude that the algorithm, using the default settings, converges on about the same minimum as the significantly longer-running one.[5]

To test how our network ranking compares to other rankings, we correlate it with the Handelsblatt (HB) 2019 ranking and the Times Higher Education (THE) 2020 ranking in Economics & Econometrics.[6] The HB ranking is considered the most important ranking to measure the research productivity of individuals and institutions in AT/CH/DE. It is based on quality-weighted publications and aggregated at the institution level (for the methodology, see [18]; for criticism of the weights used, see [19]). The THE ranking measures the reputation of institutions and has a more general approach than the HB ranking. To measure reputation, it attributes weights to different performance indicators (some of which are in part collected in a reputation survey) from five areas: teaching, research, citations, international outlook and industry income (for more details, see [20]). Table 3 shows the correlation matrix of the three rankings. The network ranking correlates positively and significantly with the THE ranking. (Note the small number of correlated observations: the HB ranking publishes only the 25 highest-ranked institutions; the THE ranking only has 28 included institutions with an *exact* rank.)

[3]Sparsity describes the fact that an adjacency matrix has a high number of entries with zeros. This is the case for our network data: we we only consider 552 individuals in a matrix with 10 000 cells.

[4]For details on the code, see [9], the electronic supplementary material of [9], and the authors' documentation of the MATLAB code.

[5]Note that each time the algorithm is run (on the same dataset with the same settings), it calculates slightly different percentages of violations. After all, it is a stochastic algorithm with random pairs of institutions both during burn-in and the sampling phase.

[6]Both rankings are also shown in electronic supplementary material, table A1.

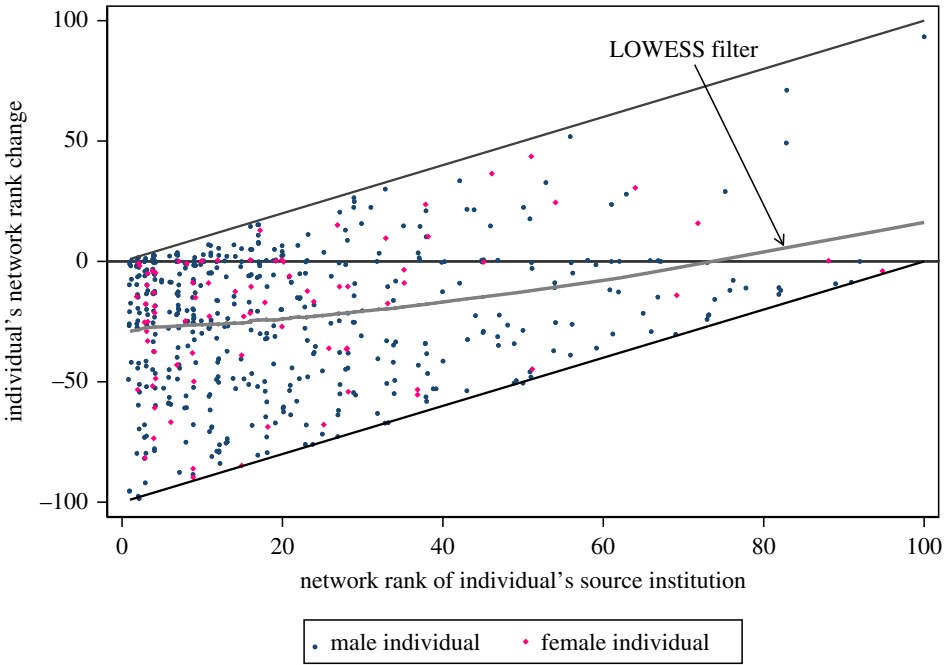

**Figure 3.** Scatter-plot of the rank changes ( jittered) with LOWESS filter.

**Table 3.** Correlation matrix of the three rankings. (Note: Kendall rank correlation coefficients, adjusted for ties.)

|  | network ranking | HB ranking | THE ranking |
|---|---|---|---|
| network ranking | 1.000 |  |  |
|  | [$n = 100$] |  |  |
| HB ranking | 0.2200 | 1.000 |  |
|  | ($p = 0.1290$) |  |  |
|  | [$n = 25$] | [$n = 25$] |  |
| THE ranking | 0.3351 | 0.3410 | 1.000 |
|  | ($p = 0.0134$) | ($p = 0.0532$) |  |
|  | [$n = 28$] | [$n = 18$] | [$n = 28$] |

# 3. Results

Now we turn to the individuals' rank changes based on the network ranking. On average, an individual goes down 22.6 ranks (s.d.=28.3, median=18) from his/her source institution to his/her target institution. This is significantly different from zero (two-sided $t$-test, $p < 0.001$), see the electronic supplementary material, figure A1 for a histogram of the individuals' rank changes. Figure 3 shows a scatter-plot with the rank changes of all individuals by the rank of the source institution and by gender. There, in addition, we also show a LOWESS filter [21], a non-parametric regression line that visualizes the relationship between the individuals' rank changes and the ranks of their source institutions. We observe: (i) there are more individuals from a higher-ranked source institution in the network than individuals from a lower-ranked source institution (see the electronic supplementary material, figure A2 for a histogram of the individuals' ranks of source institutions; it seems to follow a power law). (ii) There is no obvious difference between female and male individuals (an ordered logit regression shows that the difference (of −0.088) between females' and males' source institutions is not statistically different from zero; $p = 0.671$).[7]

---

[7]For these analyses, we pooled three countries. We also check the rank changes individually for each country (where institutions are located in this one country, and individuals can come from all three countries). In AT, all individuals go down on average 18.7 ranks (s.d. = 25.1, median = 8); In CH, all individuals go down on average 19.4 ranks (s.d. = 25.1, median = 10); in DE, all individuals go down

**Table 4.** Determinants of rank changes according to the network ranking. (Note: ordered logit regressions of individual rank changes. Standard errors in parentheses.)

| independent variable(s) | dependent variable: network rank changes | | |
| --- | --- | --- | --- |
| | Model 1 | Model 2 | Model 3 |
| female-*dummy* | 0.123 | 0.172 | −0.179 |
| | (0.208) | (0.211) | (0.519) |
| career age | | 0.013 | 0.010 |
| | | (0.009) | (0.010) |
| female * career age | | | 0.020 |
| | | | (0.027) |
| cutoff list | omitted | omitted | omitted |
| no. obs. | 552 | 546 | 546 |
| pseudo $R^2$ | 0.0001 | 0.0005 | 0.0006 |
| $p > \chi2$ | 0.5541 | 0.3163 | 0.4148 |

Women on average go down 20.5 ranks (s.d. = 28.7, median = 15), men 23.0 ranks (s.d. = 28.3, median = 20). Electronic supplementary material, figure A3 shows the cumulative distribution functions of rank changes by gender. The two functions lie almost exactly on top of one another. An exact two-sample Kolmogorov–Smirnov test rejects the difference of the two distributions ($p = 0.716$). Finally, table 4 shows the results of three different regression models of rank change on a gender-dummy and career age using ordered logit regressions. (The career age and an interaction term of career age with the female-dummy is included in Model 3 because females in the network are on average 3.678 years younger than males, this difference is statistically significant from zero; two-sided $t$-test, $p < 0.001$.) None of the regressions shows a significant gender hiring gap.

For a robustness check, we use the rank changes according to the HB and THE rankings. This approach has the advantage that rank changes are *exogenously* determined (i.e. are not directly based on the hiring network). However, it comes at the price of losing about 70% of observations as we cannot calculate rank changes for the *ca* 390 individuals with unranked source or target institutions (75 institutions in the HB ranking, 72 in the THE ranking). The share of female individuals in these two samples is comparable with the full and closed network (15.9% in the HB-ranked institutions and 14.9% in the THE-ranked ones). We also report the HB/THE rank changes by gender. According to the HB ranking, women on average go down 2.1 ranks (s.d. = 7.6, median = 1.5 ranks down), men 1.3 ranks (s.d. = 8.1, median = 0). The difference between men and women of about 0.8 is not statistically different from zero (two-sided t-test, $p = 0.644$). According to the THE ranking, women on average go down 3.1 ranks (s.d. = 10.7, median = 2 ranks down), men 2.0 ranks (s.d. = 8.7, median = 0). The difference between men and women of about 1.1 is not statistically different from zero (two-sided $t$-test, $p = 0.716$). We note that women step down more ranks than men when applying the HB and THE rankings, though these differences are not significant. As we lost about 70% of individuals focusing on the institutions with either an HB rank or a uniquely defined THE rank, these estimates are likely to be noisy. We report scatter-plots and regressions in the electronic supplementary material. The scatter-plots in electronic supplementary material, figures A4 and A5 give a qualitatively similar picture to figure 3. We note that the sign of the gender-*dummy* in all specifications in electronic supplementary material tables A2 and A3 is negative, in contrast to the coefficient's signs in the regressions shown in table 4. However, all coefficients in all regressions are not statistically significant from zero.

So, while using the three different rankings, we do not observe a significant gender hiring gap, it is interesting to note that the *levels* of average rank changes are very different between the three rankings. Using the HB ranking, all individuals go down on average 1.5 ranks (s.d. = 8.0, median = 0); with the THE ranking all individuals go down on average 2.1 ranks (s.d. = 9.0, median = 0). This is in stark contrast to the 22.6 ranks (s.d. = 28.3, median = 18), reported at the beginning of this section, when we apply our

on average 23.5 ranks (s.d. = 29.1, median = 20). In none of the three countries, two-sided *t*-tests detect a difference between male and female individuals that is significant at all conventional levels.

network data. Three points might help explain the high level of rank changes in the network ranking. First, the network ranking reflects the idea that institutions *only* consider the individuals' PhD granting institution when hiring a professor. This idea has some appeal, but it might seem a bit over-simplistic. Many other factors can also play an important role in hiring an individual: e.g. the individual's research output (for which the PhD granting institution might not necessarily be an accurate predictor), acquired third-party funding, relevant teaching experience, the individual's fit into the department, among other factors. Second, in our closed network with three countries, we drop individuals who attained their PhD at institutions outside these countries with a potentially very high prestige/reputation. This might lead to a misclassification of institutions. Third, the algorithm—by design—maximizes the average number of ranks an individual steps down. The ranking was designed to rank those institutions highest that supply the most students to 'lower ranked' institutions (remember that, in the initial step of the algorithm, institutions are ordered by their out-degree) while also minimizing the extent to which 'lower ranked' institutions supply students to them in return (the algorithm randomly chooses pairs of institutions to minimize the number of violations). Thus, the rank changes according to the network ranking represent the upper bound of rank changes.

# 4. Discussion

Let us discuss some caveats and limitations of our results, and some lessons we can learn from them for future research. First, the network ranking is a result of a static snapshot of stock data in April 2020. While some individuals changed their employing institution (potentially various times), others did not (and might be at their institution for a long time). A dynamic ranking that takes all transactions into account (giving higher weight to more recent transactions) might be more suitable. But it is very difficult to construct a dynamic dataset (or time-varying network) with all the individuals' transactions. We focused on publicly available data, and most variables beyond the PhD date are hard to find or construct.

Second, the high share of hired faculty from outside the closed network in Switzerland might influence the ranking. If Swiss institutions not only hire from outside of AT/CH/DE but also produce for this more international market, this might bias the ranking. The fact that the network ranking does not significantly correlate with the Handelsblatt ranking might be explained by both the static data aspect mentioned before, and the higher degree of internationalization of Swiss institutions (the relatively low performance of Swiss institutions in our ranking compared to the Handelsblatt ranking points into this direction).

Third, the fact that we do not observe a gender hiring gap does not mean that there is not one, controlling for more variables (e.g. institutional and individual characteristics and the interaction of the two sets of variables).[8] On the one hand, future work should include a wider range of control variables. On the other hand, more observations (e.g. including other countries' institutions) could approach the problem that economics is a field with a low share of female faculty and that maybe more (female) observations are needed to detect a gender hiring gap.

Fourth, one might object that the low share of female economists in our network might have an effect on the ranking (which is based on observed transactions). However, we do not observe a difference in the rank of the females' source institution in comparison to their male colleagues. Relatedly, the share of females in our network is comparable to the share of female faculty of computer scientists in [9] (where females also account for only 15% of faculty) and a gender hiring gap is detected, however with a higher number of observations than in our study.

Fifth, we only observe the individuals who achieved a tenured position in an economics department in AT/CH/DE. We do not have information about the pool of individuals who graduate with a Ph.D. from the institutions we consider and who do not achieve tenure (conditional on trying). Given the 'leaky pipeline', the problem that the share of women decreases at each academic career step [5,6], it seems very relevant to examine the combination of (i) the gender difference of attaining a tenured position or not (the so-called extensive margin in labour economics) and (ii) the gender hiring gap (the so-called intensive margin), using a minimum violation ranking.[9] However, the data supplied by

---

[8]The insight that the composition of search/tenure commissions might matter for the evaluation of candidates [4] and our observation that exchange of individuals between institutions of different countries is not pronounced point into this direction. The target institutions' gender composition could be included to control for the former effect. Both the source and target institutions geographical references could be used to calculate distances to further examine the latter observation.

[9]For a good distinction of the extensive and the intensive margin in another context, see [22].

universities about their PhD graduates is far from complete (for privacy and data protection reasons). Future work could try to address this issue by working with (anonymized) data from a centralized academic job market (like the European Job Market of Economists), which has recently been introduced in Europe.

Sixth, it seems important to conduct more tests with exogenously determined rankings. We worked with data from two very relevant rankings and had to drop many of the individuals as only about a quarter of institutions in our data-set were ranked there. More complete rankings (e.g. beyond the top 25 institutions) would help for either robustness checks in studies using minimum violation rankings or as an exploratory variable when using different approaches than minimum violation rankings.

Ethics. Our dataset only consists of publicly available data. We do not identify individuals.

Data accessibility. Additional tables and figures are provided in electronic supplementary material [23]. The datasets and codes generated and analysed for the current study are available in an Open Science Framework repository, https://osf.io/usfy2/.

Authors' contributions. A.O.: conceptualization, formal analysis, investigation, project administration, supervision, visualization, writing—review and editing; M.P.: data curation, formal analysis, investigation, visualization, writing—original draft.

Competing interests. The authors declare no competing interests.

Funding. For conducting their research, the authors have not received any funding from any agency. To cover publication fees, the authors acknowledge the support of Deutsche Forschungsgemeinschaft (German Research Foundation), Projektnummer 491466077, and the Open Access Publication Fund of University of Potsdam.

Acknowledgements. We thank Aaron Clauset for providing the code for the ranking. We thank the participants of research seminars at University of Potsdam and the 2nd AMSE Summer School for valuable comments and Birte Röttges for excellent research assistance. Three anonymous reviewers' helpful criticism led to a greatly improved paper.

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
