## [Peer Review File · Royal Society Open Science]

Review History

RSOS-210717.R0 (Original submission)

Review form: Reviewer 1

Is the manuscript scientifically sound in its present form?

No

Are the interpretations and conclusions justified by the results?

No

Is the language acceptable?

No

Do you have any ethical concerns with this paper?

No

Have you any concerns about statistical analyses in this paper?

Yes

Recommendation?

Reject

Comments to the Author(s)

NA

Review form: Reviewer 2**Is the manuscript scientifically sound in its present form?**

Yes

Are the interpretations and conclusions justified by the results?

Yes

Is the language acceptable?

Yes

Do you have any ethical concerns with this paper?

No

Have you any concerns about statistical analyses in this paper?

Yes

Recommendation?

Accept with minor revision (please list in comments)

Comments to the Author(s)

Summary

The authors use a minimum violations ranking method to rank academic institutions that grant PhDs in economics in Austria, Switzerland and Germany. Based on this ranking, they find that the average Austrian/Swiss/German academic obtains employment at an institution ranked 22.6 places below the institution where he obtained his Ph.D. Women, on average, go down 20.5 ranks; men on average go down 23.0 ranks. The difference between genders is not significant.

Recommendation

This is a simple application of a study by Clauset, Arbesman and Larremore (2019) in a slightly different context. (Clauset et al. investigated business, computer science and history at academic institutions in the US and Canada.) I am happy to recommend acceptance but ask that the authors be a bit clearer about the limitations of their study (see below).

Main comments

1. A minimum violations ranking (MVR) method, as its name suggests, ranks objects/individuals/whatever so as to minimise the possible number of violations of some predetermined "objective" criteria. In this case, the ranking criterion is to have as few cases as possible where university i hires an individual with a PhD from university j and university i is ranked above university j . For example, suppose there were only two institutions, Zürich ETH and St Gallen. If Zürich ETH hired 10 people from St Gallen and St Gallen hired 5 people from Zürich ETH, then St Gallen would be ranked above Zürich ETH.

Like all rankings, this ranking potentially suffers from non-classical measurement error.

* First, of course, PhD placement is not necessarily an accurate predictor of research quality (or whatever objective the ranking is meant to capture). (The authors indirectly recognise this critique by showing that their ranking highly correlates with the THE and HB rankings.)

* But second, even if it were, there are ways in which this indicator would still misclassify institutions, particularly in a closed network. For example, what if Zürich ETH hired 10 people from St Gallen, 50 people from MIT and no people from the University of Salford while St Gallen hired 5 people from Zürich ETH, no people from MIT and 55 people from the University of Salford? Then in a closed network that excluded hires from MIT and the University of Salford, St Gallen would be ranked above Zürich ETH even though most people would conclude (based on this fictional hiring data) that St Gallen should probably be ranked above Zürich ETH.

* Finally, given the indicator's design, *of course* individuals will fall in rank from their doctoral institutions. The ranking was designed to rank highest those institutions that supply the most students to "lower ranked" institutions while also minimising the extent to which "lower ranked" institutions supply students to them in return. Thus, this indicator almost assuredly overestimates the extent to which the average faculty "step down" ranks from their doctoral institution to their employing institution.

I downloaded the authors' data to try to understand the extent to which these last two points may bias their estimate that, on average, faculty step down 23 ranks from their doctoral institution to their employing institution. It matters a lot! I found (as the authors did) that the mean rank change among in-network academics (*i.e.*, they obtained their PhD and were employed by one of the included German, Swiss or Austrian institutions) using the MVR was -22.60 (s.e. 1.20). When the Handelsblatt or THE rankings were used, however, the average fall in rankings was only -2.13 ranks (Handelsblatt) and -1.45 (Times Higher Education). Thus, yes, it does appear that individuals tend to (on average) be hired by institutions ranked lower than their PhD institution but that fall is probably not as steep as the MVR would suggest.

I would therefore ask the authors to show average rank changes using all three rankings and report them in both the body and the abstract of their paper. I would also ask them to provide a more thorough explanation of the MVR as well as more detailed discussion of its limitations. (And, particularly the problems involved when using an MVR ranking to estimate X when X is related to the criterion used to determine the MVR ranking to begin with.)

2. The number of in-network women is small. In the main MVR results, the authors find that in-network women (81 individuals) fall -20.46 ranks whereas in-network men fall -23.0 ranks. (So men fall more than women.) In the THE and HB results, however, they find that in-network women fall -3.1 (24 individuals) and -2.1 ranks (26 individuals), respectively, whereas men fall -2.0 and -1.3 ranks, respectively. (So women fall more than men.) None of the three differences is statistically different from zero.

Anyway, it looks like there isn't much of a gender difference in rank changes in any of the indicators, but given the problems already identified with using the MVR indicator and the small numbers of women available in the THE and HB samples, I'd be more hesitant about drawing the conclusion that there's *no* hiring gap (regardless of direction). I would therefore ask you to discuss the THE and HB results in the body of the paper as well as in the appendix. If possible, please also highlight in the abstract and introduction that, while you do not observe a statistically significant gender hiring gap, your estimates are noisy and this is likely due to the small number of in-network women in the sample.

Minor comments

1. Could you list the THE and HB rankings of in-network institutions in the appendix?
2. What is considered a "non-economics" department? Could you give specific examples?
3. Typo in the final paragraph on p. 4: capitalise "however".

Decision letter (RSOS-210717.R0)

Dear Dr Orland

The Editors assigned to your paper RSOS-210717 "Is there a gender hiring gap in academic economics? Evidence from a network analysis" have now received comments from reviewers and would like you to revise the paper in accordance with the reviewer comments and any comments from the Editors. Please note this decision does not guarantee eventual acceptance.

Please submit your revised manuscript and required files (see below) no later than 21 days from today's (ie 24-Aug-2021) date. Note: the ScholarOne system will 'lock' if submission of the revision is attempted 21 or more days after the deadline. If you do not think you will be able to meet this deadline please contact the editorial office immediately.

on behalf of Prof Mark Chaplain (Subject Editor)
openscience@royalsociety.org

Associate Editor Comments to Author:

Thanks for your patience, while we sought reviewer commentary on your paper. Two reviewers have expressed a number of concerns, comments and queries that you will be required to respond to and satisfy the reviewers that the work has reached a publishable standard.

As they were communicated to the Editor directly, we here provide the substantive comments of reviewer 1:

==REVIEWER COMMENTS==

About the current draft, I have several concerns related to it.

(1) Doubtful ranking metrics:

The current manuscript used a similar approach with [1], but inaccurately. For example, the authors mentioned that they applied the Minimum Violation Ranking [2] to measure the ranks of institutions by 1000 repetitions of each run (p. 4). Based on the description, it is unclear whether the system converged on the minimum number of violations or not. Since their current results are based on the institutional ranking derived from the iterations, I am unsure about their rankings.

(2) No explanation why they integrated three countries' hiring datasets:

In the current draft, the authors integrated three countries' (Austria, Germany, and Switzerland) economic faculty data without proposing a reason. When we check the summary of the dataset (Table 1 in the manuscript), we can see there are only 8 institutions of Austria and 10 institutions of Switzerland among 100 institutions. Considering the small number of institutions in these countries, measuring rankings of them with other countries' institutions (such as 80 institutions of Germany) might bias the ranking results. They should have checked if the inclusion can make a bias in their results, and provided evidence that the inclusion does not make biases in their results.

(3) Unclear effect of excluded data points: As the authors mentioned in the discussion section, about 44% of faculty in Switzerland were excluded because their Ph.D. granting institutions are outside of the three countries. Since the ratio is absolutely significant, we need to know the effect of the exclusion. If there are many female faculty which is related to this study's main conclusion, the results can be interpreted differently. Even though it can be a critical caveat of the study, there is no test related to this.

I think these three points are crucially unexplained in this paper by which the conclusion is unreliable.

The following points might be minor issues, but they can be also crucial to be published in your journal.

(1) Poor writing:

This study has many grammatical errors. For example, on page 4, there is a sentence "The more general THE ranking uses a set of weighted performance measures, among them a reputation survey." which I cannot understand the meaning. Also, on the same page, a sentence is not starting with a capital '(H)owever'.

In addition, they mixed some contents' categories. I mean, the explanation about the MVR should be included in the method section, but they are included in the result section. Also, in the method, they suddenly mention 'homophily' as a property of the faculty networks, and draw it in the discussion section as one of the main results. Moreover, this manuscript did not put efforts to investigate homophilic faculty hiring patterns, hence it is unclear even whether it is a result of

homophily or not. There can be many factors such as geographical constraints, cultural preferences, and family issues, and so on.

(2) Lack of information in Figures and Tables:

The descriptions of figures and tables are not sufficient to understand the result. For example, in Fig. A2, the authors just mentioned it as 'Histogram of ranks of source institutions in the closed network'. However, the actual value that the histogram is displaying is the number of faculty from a Ph.D. granting institution which is ranked by the MVR. Also, in the regression tables, it is difficult to distinguish each model.

Reviewer comments to Author:

Reviewer: 1

Comments to the Author(s)

NA

Reviewer: 2

Comments to the Author(s)

Summary

The authors use a minimum violations ranking method to rank academic institutions that grant PhDs in economics in Austria, Switzerland and Germany. Based on this ranking, they find that the average Austrian/Swiss/German academic obtains employment at an institution ranked 22.6 places below the institution where he obtained his Ph.D. Women, on average, go down 20.5 ranks; men on average go down 23.0 ranks. The difference between genders is not significant.

Recommendation

This is a simple application of a study by Clauset, Arbesman and Larremore (2019) in a slightly different context. (Clauset et al. investigated business, computer science and history at academic institutions in the US and Canada.) I am happy to recommend acceptance but ask that the authors be a bit clearer about the limitations of their study (see below).

Main comments

1. A minimum violations ranking (MVR) method, as its name suggests, ranks objects/individuals/whatever so as to minimise the possible number of violations of some predetermined "objective" criteria. In this case, the ranking criterion is to have as few cases as possible where university i hires an individual with a PhD from university j and university i is ranked above university j . For example, suppose there were only two institutions, Zürich ETH and St Gallen. If Zürich ETH hired 10 people from St Gallen and St Gallen hired 5 people from Zürich ETH, then St Gallen would be ranked above Zürich ETH.

Like all rankings, this ranking potentially suffers from non-classical measurement error.

* First, of course, PhD placement is not necessarily an accurate predictor of research quality (or whatever objective the ranking is meant to capture). (The authors indirectly recognise this critique by showing that their ranking highly correlates with the THE and HB rankings.)

* But second, even if it were, there are ways in which this indicator would still misclassify institutions, particularly in a closed network. For example, what if Zürich ETH hired 10 people from St Gallen, 50 people from MIT and no people from the University of Salford while St Gallen hired 5 people from Zürich ETH, no people from MIT and 55 people from the University of Salford? Then in a closed network that excluded hires from MIT and the University of Salford, St

Gallen would be ranked above Zürich ETH even though most people would conclude (based on this fictional hiring data) that St Gallen should probably be ranked above Zürich ETH.

* Finally, given the indicator's design, *of course* individuals will fall in rank from their doctoral institutions. The ranking was designed to rank highest those institutions that supply the most students to "lower ranked" institutions while also minimising the extent to which "lower ranked" institutions supply students to them in return. Thus, this indicator almost assuredly overestimates the extent to which the average faculty "step down" ranks from their doctoral institution to their employing institution.

I downloaded the authors' data to try to understand the extent to which these last two points may bias their estimate that, on average, faculty step down 23 ranks from their doctoral institution to their employing institution. It matters a lot! I found (as the authors did) that the mean rank change among in-network academics (*i.e.*, they obtained their PhD and were employed by one of the included German, Swiss or Austrian institutions) using the MVR was -22.60 (s.e. 1.20). When the Handelsblatt or THE rankings were used, however, the average fall in rankings was only -2.13 ranks (Handelsblatt) and -1.45 (Times Higher Education). Thus, yes, it does appear that individuals tend to (on average) be hired by institutions ranked lower than their PhD institution but that fall is probably not as steep as the MVR would suggest.

I would therefore ask the authors to show average rank changes using all three rankings and report them in both the body and the abstract of their paper. I would also ask them to provide a more thorough explanation of the MVR as well as more detailed discussion of its limitations. (And, particularly the problems involved when using an MVR ranking to estimate X when X is related to the criterion used to determine the MVR ranking to begin with.)

2. The number of in-network women is small. In the main MVR results, the authors find that in-network women (81 individuals) fall -20.46 ranks whereas in-network men fall -23.0 ranks. (So men fall more than women.) In the THE and HB results, however, they find that in-network women fall -3.1 (24 individuals) and -2.1 ranks (26 individuals), respectively, whereas men fall -2.0 and -1.3 ranks, respectively. (So women fall more than men.) None of the three differences is statistically different from zero.

Anyway, it looks like there isn't much of a gender difference in rank changes in any of the indicators, but given the problems already identified with using the MVR indicator and the small numbers of women available in the THE and HB samples, I'd be more hesitant about drawing the conclusion that there's *no* hiring gap (regardless of direction). I would therefore ask you to discuss the THE and HB results in the body of the paper as well as in the appendix. If possible, please also highlight in the abstract and introduction that, while you do not observe a statistically significant gender hiring gap, your estimates are noisy and this is likely due to the small number of in-network women in the sample.

Minor comments

1. Could you list the THE and HB rankings of in-network institutions in the appendix?
2. What is considered a "non-economics" department? Could you give specific examples?
3. Typo in the final paragraph on p. 4: capitalise "however".

===PREPARING YOUR MANUSCRIPT===

one version identifying all the changes that have been made (for instance, in coloured highlight, in bold text, or tracked changes);
 a 'clean' version of the new manuscript that incorporates the changes made, but does not highlight them. This version will be used for typesetting if your manuscript is accepted.

===PREPARING YOUR REVISION IN SCHOLARONE===

- Any electronic supplementary material (ESM).
- If you are requesting a discretionary waiver for the article processing charge, the waiver form must be included at this step.
- If you are providing image files for potential cover images, please upload these at this step, and inform the editorial office you have done so. You must hold the copyright to any image provided.
- A copy of your point-by-point response to referees and Editors. This will expedite the preparation of your proof.

- Ensure that your data access statement meets the requirements at <https://royalsociety.org/journals/authors/author-guidelines/#data>. You should ensure that you cite the dataset in your reference list. If you have deposited data etc in the Dryad repository, please include both the 'For publication' link and 'For review' link at this stage.
- If you are requesting an article processing charge waiver, you must select the relevant waiver option (if requesting a discretionary waiver, the form should have been uploaded at Step 3 'File upload' above).
- If you have uploaded ESM files, please ensure you follow the guidance at <https://royalsociety.org/journals/authors/author-guidelines/#supplementary-material> to include a suitable title and informative caption. An example of appropriate titling and captioning may be found at https://figshare.com/articles/Table_S2_from_Is_there_a_trade-off_between_peak_performance_and_performance_breadth_across_temperatures_for_aerobic_scope_in_teleost_fishes_/3843624.

Author's Response to Decision Letter for (RSOS-210717.R0)

See Appendix A.

RSOS-210717.R1 (Revision)

Review form: Reviewer 2

Is the manuscript scientifically sound in its present form?

Yes

Are the interpretations and conclusions justified by the results?

Yes

Is the language acceptable?

Yes

Do you have any ethical concerns with this paper?

No

Have you any concerns about statistical analyses in this paper?

Yes

Recommendation?

Accept as is

Comments to the Author(s)

I think the authors for carefully considering my comments and am happy to recommend acceptance.

Review form: Reviewer 3

Is the manuscript scientifically sound in its present form?

Yes

Are the interpretations and conclusions justified by the results?

Yes

Is the language acceptable?

Yes

Do you have any ethical concerns with this paper?

No

Have you any concerns about statistical analyses in this paper?

No

Recommendation?

Accept with minor revision (please list in comments)

Comments to the Author(s)

See above comments

Decision letter (RSOS-210717.R1)

Dear Dr Orland

On behalf of the Editors, we are pleased to inform you that your Manuscript RSOS-210717.R1 "Is there a gender hiring gap in academic economics? Evidence from a network analysis" has been accepted for publication in Royal Society Open Science subject to minor revision in accordance with the referees' reports. Please find the referees' comments along with any feedback from the Editors below my signature.

Please submit your revised manuscript and required files (see below) no later than 7 days from today's (ie 11-Jan-2022) date. Note: the ScholarOne system will 'lock' if submission of the revision is attempted 7 or more days after the deadline. If you do not think you will be able to meet this deadline please contact the editorial office immediately.

on behalf of Mr Andrew Dunn (Associate Editor) and Mark Chaplain (Subject Editor)
openscience@royalsociety.org

Associate Editor Comments to Author:

Firstly, please accept the apologies of the journal and editorial team for the unusual delay in getting to this stage: it proved unusually difficult to be secure referees and solicit reports from them - we're grateful to the reviewers who did return reports.

Secondly, we're delighted that the reviewers are largely satisfied the paper is ready for acceptance - there are some comments from one of the referees (their comments are copied below) that we'd like you to take a look at. There are some minor comments you may like to respond to, and a number of citations that may add value to your work (and - in any case - the open review operated by the journal means the report and rebuttal will be accessible to a reader, which would seem especially valuable in this instance).

Otherwise, congratulations on this work, and we'll look forward to receiving a final iteration in due course.

Referee report:

This analysis helps resolve an interesting debate in the literature: the field of economics stands out as an exception to gender-neutrality in tenure-track hiring (eg., in addition to the Ginther & Kahn study, Williams & Ceci, 2015 reported that male economists were less pro-female in tenure-track hiring than all other disciplinary groups and genders). While these authors employ network analysis to remove one potential source of gender bias, they acknowledge that other bias factors could still be operating. Some have claimed that hiring analyses cannot resolve the gendered findings because women may be stronger candidates than men (they survived bias during graduate training which results in only the strongest females completing their PhDs, having more publications, giving better job talks and interviews, etc.) and therefore women should be hired more often than men rather than equally often.

The authors are careful to acknowledge potential limitations that could qualify their findings, and this is laudable. However, in a few places in the manuscript, I think they are more critical of their findings than is warranted. They write: “the fact that we do not observe a gender hiring gap does not mean that there is not one, controlling for more variables (e.g. workplace and individual characteristics like number and quality of publications, acquired third-party funding, and relevant policy work could be included in future studies).” There is no evidence that women are disadvantaged by the grant review process (with the exception of a few older European analyses the data from large-scale analyses as well as meta-analyses indicates gender-fair grant reviews, e.g., Rissler et al., 2020), and there is a great deal of evidence that women publish significantly fewer articles than men and this male advantage is visible in graduate school. Recent data-scraping exercises by Huang et al and others complicates this repeated finding of male superiority in publication by showing that women publish as often as men during periods when they are working full-time but because of more career interruptions during their lifetime, women’s publications are smaller. Ceci, Ginther, Kahn & Williams (2014) found that the gender productivity gap among economists increased from 1990-1995 to 2005-2008 (from 22% to 52%). (Huang et al. did not separate out economics.) Morgan et al.’s analysis of computer science faculty revealed a penalty over women’s lifetime that is due mainly to fewer publications during the decade surrounding the birth of a child, which largely disappears later in a woman’s career. All of this is to argue that controlling for such variables as journal productivity and grant funding would be unlikely to change the hiring findings observed by these authors.

In sum, gender productivity differences are smallest in math-intensive fields (with the exception of economics), and are largest (and growing) in biology, psychology, and economics. But hiring in these fields is increasingly gender-fair, despite productivity gaps and fair grant funding and journal reviewing. This new network analysis will add a needed piece to this debate.

Ceci, S.J., Ginther, D.K., Kahn, S. & Williams, W.M. (2014). Women in academic science: A changing landscape. *Psychological Science in the Public Interest*, 15(3):75-141.

DOI:10.1177/1529100614541236

Huang, J., Gates, A. J. Sinatra, R. & Barabási, A-L. (2020). Historical comparison of gender inequality in scientific careers across countries and disciplines. *Proc. Natl. Acad. Sci. U.S.A.*, doi:10.1073/pnas.1914221117

Morgan, A.C., Way, S.F., Hoefer, M.D., Larremore, D.B., Galesic, M. & Clauset, A. (2021).

The unequal impact of parenthood in academia. *Science Advances*, 7(no. 9).

DOI: 10.1126/sciadv.abd1996

Rissler, L., Hale, K., Joffe, N. & Caruso, N. (2020). Gender differences in grant review submissions across S&A fields at the NSF (2001-2016). *Bioscience*, 9, 814-820. DOI: 10.1093/biosci/biaa072

Williams, W.M. & Ceci, S.J. (2015). National hiring experiments reveal 2: 1 faculty preference for women on STEM tenure track. *Proceedings of the National Academy of Sciences*, 112(17), 5360-5365.

Reviewer comments to Author:

Reviewer: 2

Comments to the Author(s)

I think the authors for carefully considering my comments and am happy to recommend acceptance.

Reviewer: 3

Comments to the Author(s)

See above comments

===PREPARING YOUR MANUSCRIPT===

one version should clearly identify all the changes that have been made (for instance, in coloured highlight, in bold text, or tracked changes);

===PREPARING YOUR REVISION IN SCHOLARONE===

-- Ensure that your data access statement meets the requirements at https://royalsociety.org/journals/authors/author-guidelines/#data.

You should ensure that you cite the dataset in your reference list. If you have deposited data etc in the Dryad repository, please only include the 'For publication' link at this stage. You should remove the 'For review' link.

-- If you are requesting an article processing charge waiver, you must select the relevant waiver option (if requesting a discretionary waiver, the form should have been uploaded, see 'File upload' above).

-- If you have uploaded any electronic supplementary (ESM) files, please ensure you follow the guidance at <https://royalsociety.org/journals/authors/author-guidelines/#supplementary-material> to include a suitable title and informative caption. An example of appropriate titling and captioning may be found at https://figshare.com/articles/Table_S2_from_Is_there_a_trade-off_between_peak_performance_and_performance_breadth_across_temperatures_for_aerobic_scope_in_teleost_fishes_/3843624.

Author's Response to Decision Letter for (RSOS-210717.R1)

See Appendix B.

Decision letter (RSOS-210717.R2)

Dear Dr Orland,

I am pleased to inform you that your manuscript entitled "Is there a gender hiring gap in academic economics? Evidence from a network analysis" is now accepted for publication in Royal Society Open Science.

on behalf of Prof Mark Chaplain (Subject Editor)
openscience@royalsociety.org

Appendix A

Response to the Reviewers

We thank the reviewers for their very helpful comments. Due to the comments, we think the paper improved very much by addressing them. In this response letter, you find the complete reviewer comments below, and our responses (pointing out how we addressed the comments) indented below each comment.

Reviewer 1

About the current draft, I have several concerns related to it.

(1) Doubtful ranking metrics:

The current manuscript used a similar approach with [1], but inaccurately. For example, the authors mentioned that they applied the Minimum Violation Ranking [2] to measure the ranks of institutions by 1000 repetitions of each run (p. 4). Based on the description, it is unclear whether the system converged on the minimum number of violations or not. Since their current results are based on the institutional ranking derived from the iterations, I am unsure about their rankings.

Thank you for this comment. Due to your doubt in this comment and to answer the questions raised by the other referee, we substantially revised the text regarding the MVR. You also made us aware of a mistake in understanding the MATLAB MVR code. We used the default settings of the code (with bootstrapping), and thus conducted 100 repetitions with 20,000 iterations each for the ranking. This is way more than the previously stated 36 repetitions with 1,000 iterations. We are sorry for the misunderstanding and the resulting confusion.

We also added a paragraph regarding the convergence of the algorithm (and Figure 2). We let the algorithm run significantly longer than our implementation and find that the difference between the two implementations is small. (However, we will always find differences between two runs of the algorithm as it is stochastic.)

You find the changed paragraphs on page 3, starting with "To rank the institutions..."

(2) No explanation why they integrated three countries' hiring datasets:

In the current draft, the authors integrated three countries' (Austria, Germany, and Switzerland) economic faculty data without proposing a reason. When we check the summary of the dataset (Table 1 in the manuscript), we can see there are only 8 institutions of Austria and 10 institutions of Switzerland among 100 institutions. Considering the small number of institutions in these countries, measuring rankings of them with other countries' institutions (such as 80 institutions of Germany) might bias the ranking results. They should have checked if the inclusion can make a bias in their results, and provided evidence that the inclusion does not make biases in their results.

Thank you for these comments. We agree that the motivation was rather short. We also checked the countries' gender hiring gaps individually

We discuss the reasons why we consider three countries in footnote 1 (on page 2).

In footnote 7 (page 4), we report the rank changes by country. All t-tests for gender gaps were also insignificant.

(3) Unclear effect of excluded data points: As the authors mentioned in the discussion section, about 44% of faculty in Switzerland were excluded because their Ph.D. granting institutions are outside of the three countries. Since the ratio is absolutely significant, we need to know the effect of the

exclusion. If there are many female faculty which is related to this study's main conclusion, the results can be interpreted differently. Even though it can be a critical caveat of the study, there is no test related to this.

Thank you for this very helpful comment. It is true, we have to check if included and excluded individuals are different. To test if the gender ratio of the included economists in the closed network differs from the excluded economists, we show a two-way tabulation in new Table 2 (page 6), both for our total sample, and for each of the three countries.

When we consider the row percentages, we see that the share of females in the closed network is close to the excluded individuals. This is confirmed by Fisher's exact tests for both our total sample, and for the three countries individually. (The results of chi-square tests are qualitatively similar. We report only Fisher's exact tests in the paper as some cells have a quite small number of observations.)

We conclude that there is no significant effect from excluding Ph.D.s from outside the closed network on the gender composition in the closed network.

You can find this discussion on page 3. The paragraph starts with "The restriction..."

I think these three points are crucially unexplained in this paper by which the conclusion is unreliable. The following points might be minor issues, but they can be also crucial to be published in your journal.

(1) Poor writing:

This study has many grammatical errors. For example, on page 4, there is a sentence "The more general THE ranking uses a set of weighted performance measures, among them a reputation survey." which I cannot understand the meaning. Also, on the same page, a sentence is not starting with a capital '(H)owever'.

Thank you for this comment. We gave the paper another careful read and revised it carefully. Also two different proof-reading software packages could not find other grammatical errors.

Now we explain the THE ranking in more detail. (We think we previously tried to explain too much in one single sentence.) You can find this new explanation on page 4 (paragraph starting with "To test how...").

In addition, they mixed some contents' categories. I mean, the explanation about the MVR should be included in the method section, but they are included in the result section. Also, in the method, they suddenly mention 'homophily' as a property of the faculty networks, and draw it in the discussion section as one of the main results. Moreover, this manuscript did not put efforts to investigate homophilic faculty hiring patterns, hence it is unclear even whether it is a result of homophily or not. There can be many factors such as geographical constraints, cultural preferences, and family issues, and so on.

Thank you for this comment. We moved the explanation regarding the minimum violation ranking to the methods section and substantially extended it.

It is true that the observation of homophily is not a key finding of this paper (maybe surprising nonetheless). As we do not have further variables to examine this observation, we attenuate the sentence (when talking about Figure 1) and deleted our list of "takeaway messages" at the end of the paper.

(2) Lack of information in Figures and Tables:

The descriptions of figures and tables are not sufficient to understand the result. For example, in Fig. A2, the authors just mentioned it as 'Histogram of ranks of source institutions in the closed network'. However, the actual value that the histogram is displaying is the number of faculty from a Ph.D. granting institution which is ranked by the MVR. Also, in the regression tables, it is difficult to distinguish each model.

Thank you for this comment. We had a critical look at all tables and figures and extended them by additional information (also in the captions). We also explain the LOWESS filter in the text now (page 4, first paragraph of the results section). We hope this increases the ease of understanding.

Reviewer 2

Summary

The authors use a minimum violations ranking method to rank academic institutions that grant PhDs in economics in Austria, Switzerland and Germany. Based on this ranking, they find that the average Austrian/Swiss/German academic obtains employment at an institution ranked 22.6 places below the institution where he obtained his Ph.D. Women, on average, go down 20.5 ranks; men on average go down 23.0 ranks. The difference between genders is not significant.

Recommendation

This is a simple application of a study by Clauset, Arbesman and Larremore (2019) in a slightly different context. (Clauset et al. investigated business, computer science and history at academic institutions in the US and Canada.) I am happy to recommend acceptance but ask that the authors be a bit clearer about the limitations of their study (see below).

Main comments

1. A minimum violations ranking (MVR) method, as its name suggests, ranks objects/individuals/whatever so as to minimise the possible number of violations of some predetermined "objective" criteria. In this case, the ranking criterion is to have as few cases as possible where university *i* hires an individual with a PhD from university *j* and university *i* is ranked above university *j*. For example, suppose there were only two institutions, Zürich ETH and St Gallen. If Zürich ETH hired 10 people from St Gallen and St Gallen hired 5 people from Zürich ETH, then St Gallen would be ranked above Zürich ETH.

Like all rankings, this ranking potentially suffers from non-classical measurement error.

* First, of course, PhD placement is not necessarily an accurate predictor of research quality (or whatever objective the ranking is meant to capture). (The authors indirectly recognise this critique by showing that their ranking highly correlates with the THE and HB rankings.)

* But second, even if it were, there are ways in which this indicator would still misclassify institutions, particularly in a closed network. For example, what if Zürich ETH hired 10 people from St Gallen, 50 people from MIT and no people from the University of Salford while St Gallen hired 5 people from Zürich ETH, no people from MIT and 55 people from the University of Salford? Then in a closed network that excluded hires from MIT and the University of Salford, St Gallen would be ranked above Zürich ETH even though most people would conclude (based on this fictional hiring data) that St Gallen should probably be ranked above Zürich ETH.

* Finally, given the indicator's design, *of course* individuals will fall in rank from their doctoral institutions. The ranking was designed to rank highest those institutions that supply the most students to "lower ranked" institutions while also minimising the extent to which "lower ranked" institutions supply students to them in return. Thus, this indicator almost assuredly overestimates the extent to which the average faculty "step down" ranks from their doctoral institution to their employing institution.

I downloaded the authors' data to try to understand the extent to which these last two points may bias their estimate that, on average, faculty step down 23 ranks from their doctoral institution to their employing institution. It matters a lot! I found (as the authors did) that the mean rank change among in-network academics (*i.e.*, they obtained their PhD and were employed by one of the included German, Swiss or Austrian institutions) using the MVR was -22.60 (s.e. 1.20). When the

Handelsblatt or THE rankings were used, however, the average fall in rankings was only -2.13 ranks (Handelsblatt) and -1.45 (Times Higher Education). Thus, yes, it does appear that individuals tend to (on average) be hired by institutions ranked lower than their PhD institution but that fall is probably not as steep as the MVR would suggest.

I would therefore ask the authors to show average rank changes using all three rankings and report them in both the body and the abstract of their paper. I would also ask them to provide a more thorough explanation of the MVR as well as more detailed discussion of its limitations. (And, particularly the problems involved when using an MVR ranking to estimate X when X is related to the criterion used to determine the MVR ranking to begin with.)

Thank you very much for this comment. First of all, now, we discuss the mechanics of the algorithm underlying the MVR in more detail in the methods section (page 3 and 4, paragraph starting with "To rank..."). We also mention the average rank changes according to the HB and THE ranking at the end of the results section and compare them with the ones from the network ranking (page 5).

We take this finding as a starting point to reflect on your points of criticism. First of all, we point out that the network ranking reflects the idea that institutions *only* consider the individuals' Ph.D. granting institution when hiring a professor. This idea has some appeal, but it might seem a bit oversimplistic. Many other factors can also play an important role for hiring a person: the individuals' research output, acquired third-party funding, teaching experience, the individual's fit into the department, etc. (I.e. your argument that Ph.D. placement is not necessarily an accurate predictor of research quality.) Then, we discuss the problem of a closed network in our case. We drop many individuals with Ph.D.s from universities outside of the three countries which have a very high reputation. This might indeed lead to a misclassification of institutions. Finally, we discuss the problem that when applying the MVR algorithm, this entails that the average drop in ranks per individual is maximised (and that, by design, the number of ranks that each individual steps down, is overestimated). (You find this in the last paragraph of the results section.)

2. The number of in-network women is small. In the main MVR results, the authors find that in-network women (81 individuals) fall -20.46 ranks whereas in-network men fall -23.0 ranks. (So men fall more than women.) In the THE and HB results, however, they find that in-network women fall -3.1 (24 individuals) and -2.1 ranks (26 individuals), respectively, whereas men fall -2.0 and -1.3 ranks, respectively. (So women fall more than men.) None of the three differences is statistically different from zero.

Anyway, it looks like there isn't much of a gender difference in rank changes in any of the indicators, but given the problems already identified with using the MVR indicator and the small numbers of women available in the THE and HB samples, I'd be more hesitant about drawing the conclusion that there's **no** hiring gap (regardless of direction). I would therefore ask you to discuss the THE and HB results in the body of the paper as well as in the appendix. If possible, please also highlight in the abstract and introduction that, while you do not observe a statistically significant gender hiring gap, your estimates are noisy and this is likely due to the small number of in-network women in the sample.

Thank you for this comment. We now show the rank changes by gender in the main text of the last paragraph of the results section..

We also discuss the problem that when considering the HB and THE ranking, we reduce data-set by so many observations that it becomes hard to present statistically sound evidence. We

also discuss the consequences of this comment (and your previous one) in the discussion section.

Minor comments

1. Could you list the THE and HB rankings of in-network institutions in the appendix?

Thank you for the comment. Table A.1 now also lists the HB and THE ranks of the institutions.

2. What is considered a "non-economics" department? Could you give specific examples?

Thanks. There are a number of related fields to economics. Examples are statistics (as a method applied in economics), business, political science, or agricultural economics. We mention these fields now when restricting the individuals' Ph.D. to economics and refer to them when we restrict the departments to economics departments.

3. Typo in the final paragraph on p. 4: capitalise "however".

Thanks. We fixed it.

Appendix B

Response to the Reviewer

We thank the reviewer of our resubmission for the very helpful comments. We think that this perspective greatly improves the discussion of the paper. In this response letter, you find the complete reviewer comment below, and our response (pointing out how we addressed the comments) indented below the comment.

Reviewer

This analysis helps resolve an interesting debate in the literature: the field of economics stands out as an exception to gender-neutrality in tenure-track hiring (eg., in addition to the Ginther & Kahn study, Williams & Ceci, 2015 reported that male economists were less pro-female in tenure-track hiring than all other disciplinary groups and genders). While these authors employ network analysis to remove one potential source of gender bias, they acknowledge that other bias factors could still be operating. Some have claimed that hiring analyses cannot resolve the gendered findings because women may be stronger candidates than men (they survived bias during graduate training which results in only the strongest females completing their PhDs, having more publications, giving better job talks and interviews, etc.) and therefore women should be hired more often than men rather than equally often.

The authors are careful to acknowledge potential limitations that could qualify their findings, and this is laudable. However, in a few places in the manuscript, I think they are more critical of their findings than is warranted. They write: “the fact that we do not observe a gender hiring gap does not mean that there is not one, controlling for more variables (e.g. workplace and individual characteristics like number and quality of publications, acquired third-party funding, and relevant policy work could be included in future studies).” There is no evidence that women are disadvantaged by the grant review process (with the exception of a few older European analyses the data from large-scale analyses as well as meta-analyses indicates gender-fair grant reviews, e.g., Rissler et al., 2020), and there is a great deal of evidence that women publish significantly fewer articles than men and this male advantage is visible in graduate school. Recent data-scraping exercises by Huang et al and others complicates this repeated finding of male superiority in publication by showing that women publish as often as men during periods when they are working full-time but because of more career interruptions during their lifetime, women’s publications are smaller. Ceci, Ginther, Kahn & Williams (2014) found that the gender productivity gap among economists increased from 1990-1995 to 2005-2008 (from 22% to 52%). (Huang et al. did not separate out economics.) Morgan et al.’s analysis of computer science faculty revealed a penalty over women’s lifetime that is due mainly to fewer publications during the decade surrounding the birth of a child, which largely disappears later in a woman’s career. All of this is to argue that controlling for such variables as journal productivity and grant funding would be unlikely to change the hiring findings observed by these authors.

In sum, gender productivity differences are smallest in math-intensive fields (with the exception of economics), and are largest (and growing) in biology, psychology, and economics. But hiring in these fields is increasingly gender-fair, despite productivity gaps and fair grant funding and journal reviewing. This new network analysis will add a needed piece to this debate.

Ceci, S.J., Ginther, D.K., Kahn, S. & Williams, W.M. (2014). Women in academic science: A changing landscape. *Psychological Science in the Public Interest*, 15(3):75-141.

DOI:10.1177/1529100614541236

Huang, J., Gates, A. J. Sinatra, R. & Barabási, A-L. (2020). Historical comparison of gender inequality in scientific careers across countries and disciplines. *Proc. Natl. Acad. Sci. U.S.A.*, doi:10.1073/pnas.1914221117

Morgan, A.C., Way, S.F., Hoefler, M.D., Larremore, D.B., Galesic, M. & Clauset, A. (2021). The unequal impact of parenthood in academia. *Science Advances*, 7(no. 9). DOI: 10.1126/sciadv.abd1996

Rissler, L., Hale, K., Joffe, N. & Caruso, N. (2020). Gender differences in grant review submissions across S&A fields at the NSF (2001-2016). *Bioscience*, 9, 814-820. DOI: 10.1093/biosci/biaa072

Williams, W.M. & Ceci, S.J. (2015). National hiring experiments reveal 2: 1 faculty preference for women on STEM tenure track. *Proceedings of the National Academy of Sciences*, 112(17), 5360-5365.

Thank you very much for the comments. We think that they contribute to a more evenly balanced paper.

We now mention the interesting findings by Williams & Ceci (2015) in the first paragraph of the introduction (p. 2).

Your criticism in the second paragraph is fair. We made this part of the discussion clearer, using the Williams & Ceci (2015) reference. (p. 10).